# Marginal ice zone fraction benchmarks sea ice and climate model skill

Christopher Horvat [1✉]

Global climate models (GCMs) consistently underestimate the response of September Arctic sea-ice area (SIA) to warming. Modeled SIA losses are highly correlated to global mean temperature increases, making it challenging to gauge if improvements in modeled sea ice derive from improved sea-ice models or from improvements in forcing driven by other GCM components. I use a set of five large GCM ensembles, and CMIP6 simulations, to quantify GCM internal variability and variability between GCMs from 1979–2014, showing modern GCMs do not plausibly estimate the response of SIA to warming in all months. I identify the marginal ice zone fraction (MIZF) as a metric that is less correlated to warming, has a response plausibly simulated from January–September (but not October–December), and has highly variable future projections across GCMs. These qualities make MIZF useful for evaluating the impact of sea-ice model changes on past, present, and projected sea-ice state.

---

[1] Brown University, Providence, RI, USA. ✉email: christopher_horvat@brown.edu

The rapid decline of the Arctic sea ice cover is a primary indicator of Earth's changing climate. Yet the response of September sea-ice area or extent (SIA/SIE) to a unit increase in carbon emissions is underestimated compared to observations in nearly all global climate models (GCMs): only models that overestimate resulting global warming can reproduce observed trends in sea ice loss[1]. Accordingly, the failure of GCMs to represent declining trends in observed SIE[2,3] has contributed to efforts to improve the representation of sea ice in GCMs[4–6]. Still, a majority of GCMs submitting to phase 6 of the Coupled Model Intercomparison Project[7] (CMIP6) remain unable to produce plausible estimates of the observed decline of September Arctic sea ice[8].

The use of SIE/SIA to evaluate sea-ice model improvements is confounded by the parametric relationship between modeled SIA and carbon emissions; and by extension global mean temperature (GMT), one that holds across sea ice and climate models of a wide range of complexities[8–13], and references within. The underestimation of September SIA loss has improved across CMIP generations, but the link between SIA and GMT makes it challenging to judge whether improvements in modeled sea ice originate in improvements in sea-ice model physics or are due primarily to improvements in the external forcing of sea ice[14], like changes to aerosol forcing[8,15–17] noted that the inaccurate representation of SIA trends in CMIP5 models can be explained as the result of inherent internal variability in modeled climate, which in this study will be referred to as "model internal variability". Because of the low number of ensemble members contributed by each model group in CMIP5, it was not possible to directly evaluate model internal variability over the global warming period. Instead,[17] used year-to-year variance from a single model realization over a long pre-industrial simulation to represent model internal variability. The recent publication of large (20+ member) ensembles[18–20], and the contribution of multiple historical runs to the 6th CMIP from a number of modeling centers (see Table 1) now allows for a direct evaluation of model internal variability for several independent models, and over the same period that sea ice has declined.

The marginal ice zone (MIZ) has been a point of focus as a critical region for polar climate and ecology[21–23], and can be observed using the same passive microwave observations used to estimate SIA/SIE[24]. MIZ location and variability is set through the coupled interaction of oceanic, atmospheric, thermodynamic and sea ice dynamical processes that are the focus of many recent modeling efforts[25–29]. However, September MIZ extent has generally been unchanged over 1979–2018[30], with September MIZ fraction (MIZF), the percentage of the September Arctic sea ice cover that is MIZ, increasing as SIA has declined.

Here I evaluate model representations of SIA, MIZF, and GMT against observations, incorporating both climate model internal variability and inter-model differences. I use five 20-50 member ensemble simulations submitted to the CLIVAR large ensemble project[20], CLIVAR-LE (Table 1, top). As mentioned above, these large (20+ member) ensembles[18–20] permit a direct assessment of model internal variability from ensemble variance. To assess inter-model differences, I create an "ensemble-mean-ensemble" (CMIP6-EME) from the ensemble-mean statistics of 8 climate models which submitted relatively large sets (10+ member) of historical simulations to CMIP6 (Table 1, bottom). These model ensembles are compared to three satellite observational products from which I derive observational uncertainty.

Across this set of 5+1 ensembles, I consider two main quantities: SIA and MIZF, and their sensitivities to warming—the relationship between a unit change in SIA/MIZF and a unit change in GMT. I find that after accounting for model internal variability, inter-model variance, and observational uncertainty, modern GCMs are generally unable to produce plausible estimates of SIA sensitivity to warming at any point during the year. On the other hand, I demonstrate that MIZF is weakly correlated to warming, and a useful benchmark for understanding GCM and sea-ice model skill. Five of six ensembles produce plausible estimates of MIZF sensitivity in 7 months or more, including September. The exception is during the fall freeze-up period from October-December, which suggests a target for improvement in current-generation sea-ice models. As future projections of September and December MIZF differ radically between models, MIZF may also be a useful way to determine in real-time whether current models make skillful predictions of future Arctic sea ice variability.

## Results

**GCM estimates of September sea ice and MIZF change.** With the largest percentage changes in Arctic sea ice coverage occurring in September[31], the ability of GCMs to reproduce September trends in sea ice coverage is paramount. Figure 1a plots September Arctic SIA for CLIVAR-LE models (colors) and the merged observational product (black), with the CMIP6-EME mean plotted as a dotted line (see methods for details on the CLIVAR-LE, how the CMIP6-EME is constructed, observational products, and how SIA, MIZF, and trends are calculated). Consistent with the model-mean behavior of the 5th and 6th

| Table 1 Global climate models used in this study. | | | | | |
|---|---|---|---|---|---|
| **Modeling center** | **Version** | **No. members** | **Sea-ice model** | **Ocean model** | **doi** |
| CLIVAR-LE | | | | | |
| CCCma | CanESM2 | 50 | CanSIM1 | CanOM4 | 10.1175/JCLI-D-16-0412.1 |
| NCAR | CESM1 | 40 | CICE4.0 | POP2 | 10.1175/BAMS-D-13-00255.1 |
| CSIRO | MK3.6 | 30 | CSIRO-SIM | MOM2.2 | 10.22499/2.6301.001 |
| GFDL | CM3 | 20 | SISp2 | MOM4.1 | 10.1175/JCLI-D-18-0134.1 |
| GFDL | ESM2M | 30 | SISp2 | MOM4.1 | 10.5194/bg-12-3301-2015 |
| CMIP6-EME | | | | | |
| NCAR | CESM2 | 11 | CICE5.1 | POP2 | 10.22033/ESGF/CMIP6.2185 |
| EC | EC-Earth3 | 14 | LIM3 | NEMO3.6 | 10.22033/ESGF/CMIP6.4700 |
| GISS | E2-1-H | 10 | GISS SI | HYCOM | 10.22033/ESGF/CMIP6.1421 |
| INM | CM5-0 | 10 | INM-ICE1 | INM-OM4 | 10.22033/ESGF/CMIP6.1423 |
| IPSL | CM6A-LR | 32 | LIM3 | OPA | 10.22033/ESGF/CMIP6.1534 |
| JAMSTEC | MIROC6 | 10 | COCO4.9 | COCO4.9 | 10.22033/ESGF/CMIP6.881 |
| MPI | ESM1-2-LR | 10 | MPI-SIM | MPIOM1.63 | 10.22033/ESGF/CMIP6.742 |
| NCC | NORCPM1 | 27 | CICE4 | MICOM1.1 | 10.22033/ESGF/CMIP6.10843 |

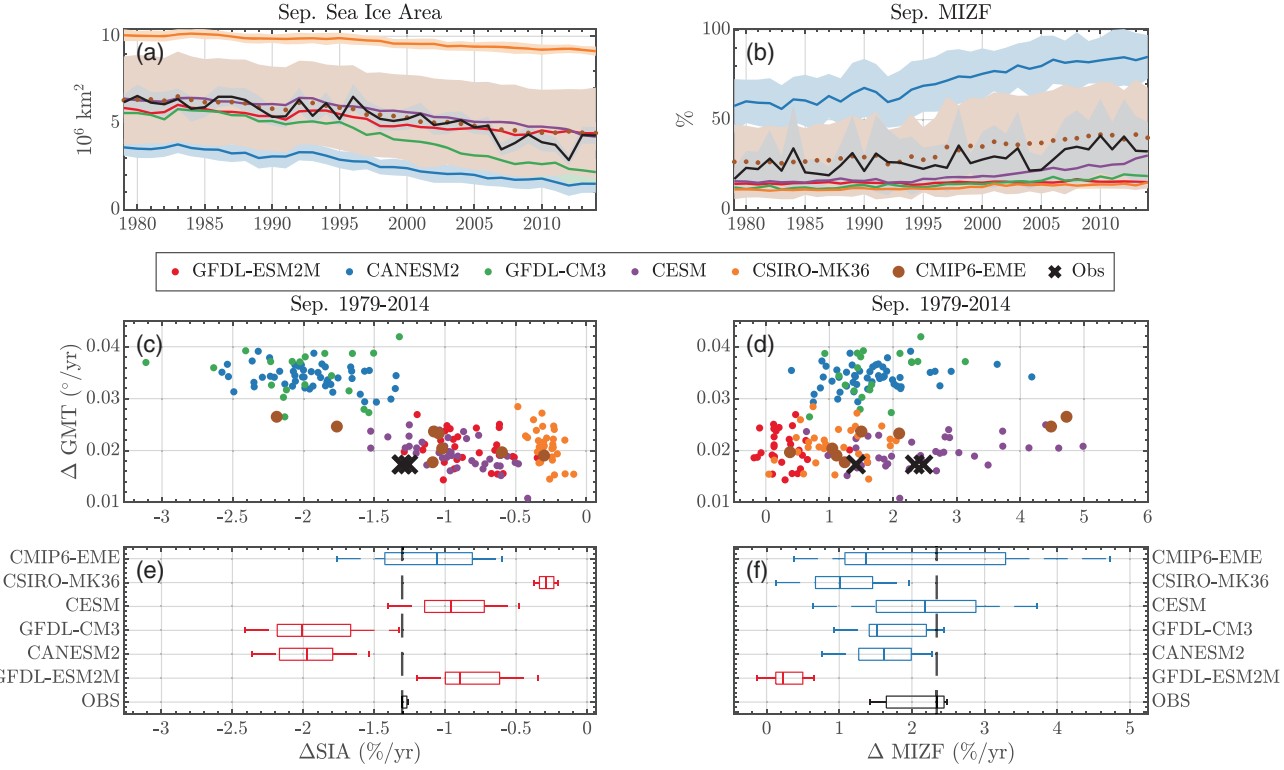

**Fig. 1 September sea ice statistics, sensitivities, and trends. a** Annually averaged SIA for all CLIVAR-LE and CMIP6-EME simulations. Solid lines are CLIVAR-LE ensemble means. Dotted line is the CMIP6-EME ensemble mean. Colored areas are one unbiased sample standard deviation from ensemble means (see "Methods"). Black lines and shaded region is the same for observations and observational uncertainty. **c** Scatter plot of 1979–2014 trend in global mean temperature vs normalized trend in sea-ice area (expressed as a percentage per year). Crosses are observations. **e** Box plots of sea-ice area trends for models shown above and observations. Whiskers extend the boxes ± 3 standard deviations from the mean in each month and vertical line is ensemble median. Dashed black line extends the median observed trend. Blue boxes are plausible estimates given the observation (black boxes, see methods), red boxes are implausible estimates (at $p = 0.01$). **b, d, f** Same as (**e**) but for MIZF.

CMIPs[3,8], while there is broad disagreement in the magnitude of SIA among individual models, the CMIP6-EME mean generally tracks observed September SIA.

For each ensemble member, a normalized SIA trend is computed ($x$ axis, $\Delta$SIA, see methods), scattered against the trend in GMT ($y$ axis, $\Delta$GMT) over the period 1979–2014. Because of the large difference between models in SIA and MIZF mean states, trends are expressed as a percentage change per year (this choice is examined in more detail in the Discussion below). While individual CLIVAR-LE models do not produce plausible estimates (see methods) of $\Delta$SIA in September (red boxes, Fig. 1e), the CMIP6-EME does (blue box, Fig. 1e), again consistent with[3,8]. I show below, as in ref. [1], that this apparent agreement is a result of an inaccurate SIA sensitivity to warming. The scatter of $\Delta$GMT vs. $\Delta$SIA in Fig. 1c illustrates their known parametric relationship[11,12,32], e.g., September $\Delta$SIA and $\Delta$GMT are significantly correlated ($p < 0.01$) with a bivariate correlation coefficient of $r = -0.81$: variation in $\Delta$GMT across all simulations can explain 66% of the variance in $\Delta$SIA.

The same analysis applied to marginal ice zone fraction (MIZF; Fig. 1b, d, f) reveals a different response. There is greater observational uncertainty in $\Delta$MIZF trends than $\Delta$SIA, as the distribution of sea ice concentration is sensitive to the algorithm used to recover ice concentration from satellites[16]. All three observational estimates show positive $\Delta$MIZF from 1979–2014. The CMIP-EME and four of five members of the CLIVAR-LE produce plausible estimates of September $\Delta$MIZF over this period.

In Fig. 2 I plot coefficients of determination (Pearson's $r^2$) between annual $\Delta$GMT and monthly $\Delta$SIA (red) and $\Delta$MIZF

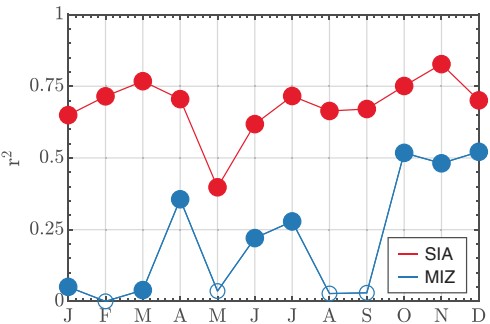

**Fig. 2 Correlation between SIA/MIZF and GMT.** Seasonal cycle of explained variance ($r^2$) of SIA trends (red) or MIZF trends (blue) by annual GMT trends across all CLIVAR-LE and CMIP6-EME simulations. Solid circles are significant correlations ($p < 0.01$).

(blue) trends, with significant correlations filled circles. SIA trends are significantly correlated with GMT trends in all months, whereas MIZF has no significant correlation ($p > 0.01$) with GMT in four: February, May, August, and September. The coefficient of determination of SIA exceeds that of MIZF in all months which can be expected as MIZF involves a ratio of MIZ area to SIA (see methods). In September, with a bivariate correlation coefficient of $r = 0.17$, only 3% of the variability in $\Delta$MIZF across the simulations can be explained by $\Delta$GMT variability.

**Seasonal GCM estimates of SIA and MIZF sensitivities.** The preceding analysis demonstrated that September MIZF trends

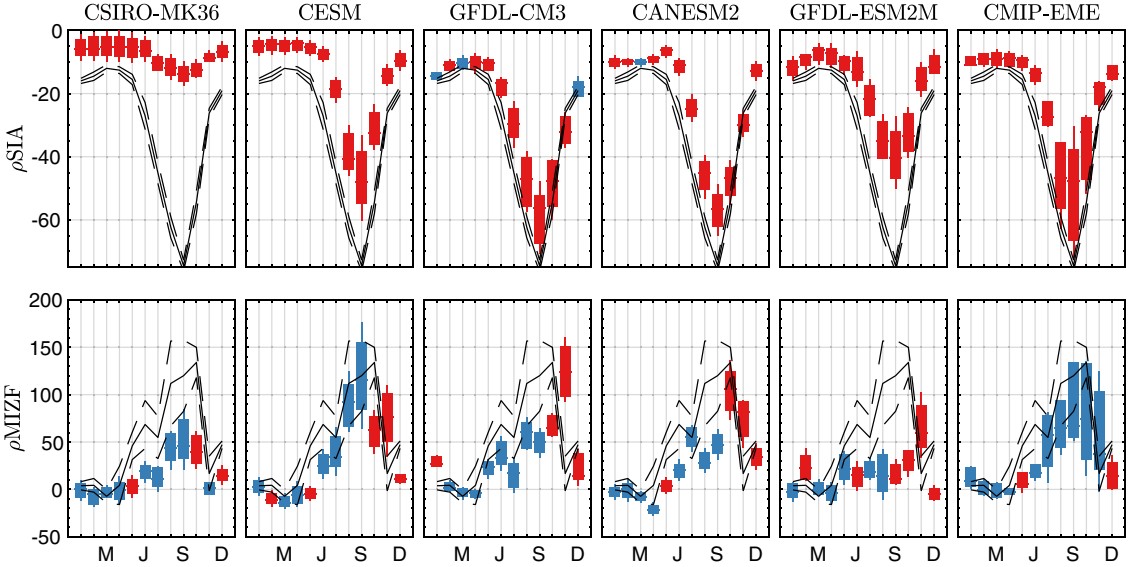

**Fig. 3 Seasonal plausibility of GCM estimates of SIA/MIZF sensitivity.** (Top row) Seasonal cycle of sea ice sensitivity $\rho$SIA (units %/K), for CLIVAR-LE members or CMIP6-EME over 1979–2014. Boxes cover the interquartile range and horizontal whiskers the ensemble median. Vertical whiskers extend to $\pm 3\sigma$ from the ensemble mean (95% of normally distributed data). Black solid line is the mean from observations. Dashed lines bound observational uncertainty (see "Methods"). Red shaded boxes are implausible estimates of sensitivity, and blue are plausible estimates (see "Methods"). (Bottom row) Same, but for MIZF sensitivity $\rho$MIZF (units %/K).

alone are plausibly estimated when considering both internal variability (in 4/5 CLIVAR-LE models) and inter-model variability (computed via the CMIP6-EME). September MIZF trends are also not correlated with GMT trends, and no members of the CLIVAR-LE plausibly estimate the observed change in September SIA. Still, the bias in modeled September SIA could plausibly be explained as a consequence of inter-model variability in the CMIP6-EME, though this does not take into account the correlation between GMT and September SIA. Here I investigate this correlation, examining SIA and MIZF sensitivity to warming across the seasonal cycle, where the sea ice sensitivity (or MIZF sensitivity), $\rho$SIA, is defined as the percent change in SIA per degree change in GMT (units %/K). A declining SIA with increasing GMT therefore yields a negative $\rho$SIA.

Figure 3 (top row) shows box plots of the seasonal cycle of sea ice sensitivity, which is defined as the percent change in sea ice extent per unit of warming ($\rho$SIA, units %/K), for each of the 5 + 1 ensembles. Note that observed global warming over 1979–2014 was ~0.634 K, and therefore $\rho$SIA = 50%/K would correspond to a −30% change in SIA over that period.

The plausibility of a given model estimate of sensitivity is evaluated via a two-sided $t$-test (see "Methods" for the rationale behind this choice). Red boxes plot where the null hypothesis— that model estimates and observational estimates come from distributions with the same mean—is rejected at the 5% significance level. Each of the ensembles reproduces the qualitative seasonal cycle of $\rho$SIA, with the largest magnitude sensitivity to warming from June to September. In all months, and across all of the ensembles, observed values of $\rho$SIA are more negative than model estimates. This includes an underestimation of September sea ice sensitivity, as previously observed by many[1,2,8], e.g., However, controlling for internal variability (in the CLIVAR-LE) or inter-model variability (in the CMIP-EME), no models make plausible estimations of September $\rho$SIA, or in more than 3 months of the year (blue boxes). This differs from the results of[8,17], who found that estimates of September sea ice sensitivity in CMIP5/6 class models were plausible when accounting for internal variability—although in those studies internal variability was quantified using different methodologies

to account for having fewer ensemble members (see "Discussion", below).

Each ensemble also qualitatively reproduces the observed seasonal cycle of MIZF sensitivity, $\rho$MIZF (bottom row, Fig. 3): increasing after January, peaking between August and October and declining to December. Accounting for internal or inter-model variability, 4/5 CLIVAR-LE members and the CMIP-EME plausibly predict MIZF sensitivity in at least 7 months. The exception is the GFDL-ESM2M model, which does in 6 months. Observational uncertainty in $\rho$MIZF is significantly higher than $\rho$SIA because of algorithmic differences between the three observational products, which may increase the plausibility of the modeled sensitivities. However, repeating the analysis for $\rho$MIZF, but synthetically using the significantly lower observational uncertainty for $\rho$SIA (see Methods, Supporting Fig. S2), yields qualitatively and quantitatively similar results, as does repeating this analysis using any two of the three observational estimates instead of all three.

Despite producing generally plausible estimates of $\rho$MIZF from January-September, all but one of the 5 CLIVAR-LEs fail to estimate $\rho$MIZF from October to December (the exception, CSIRO-MK36, provides a plausible estimate in November). These three months are those with the highest annual correlation between $\Delta$MIZF and $\Delta$GMT and therefore likely the most sensitive to forcing biases. The CMIP-EME does plausibly estimate $\rho$MIZF in October and November.

**Future projections of MIZF variability.** Accounting for internal variability, historical estimates of MIZF sensitivity are generally plausible from January to September. From October to December, internal variability cannot explain incorrect estimates of $\rho$MIZF, but inter-model variability (expressed by the CMIP-EME) may. Thus MIZF may prove a useful benchmark for characterizing future model projections of sea ice change. Figure 4 repeats Fig. 1d, f, focusing on the future (2020–2055) projections of September $\Delta$MIZF (a, c) and December $\Delta$MIZF (b, d). Only the CLIVAR-LE simulations are considered here as no 10+ member ensembles of CMIP6 data were available under the RCP8.5

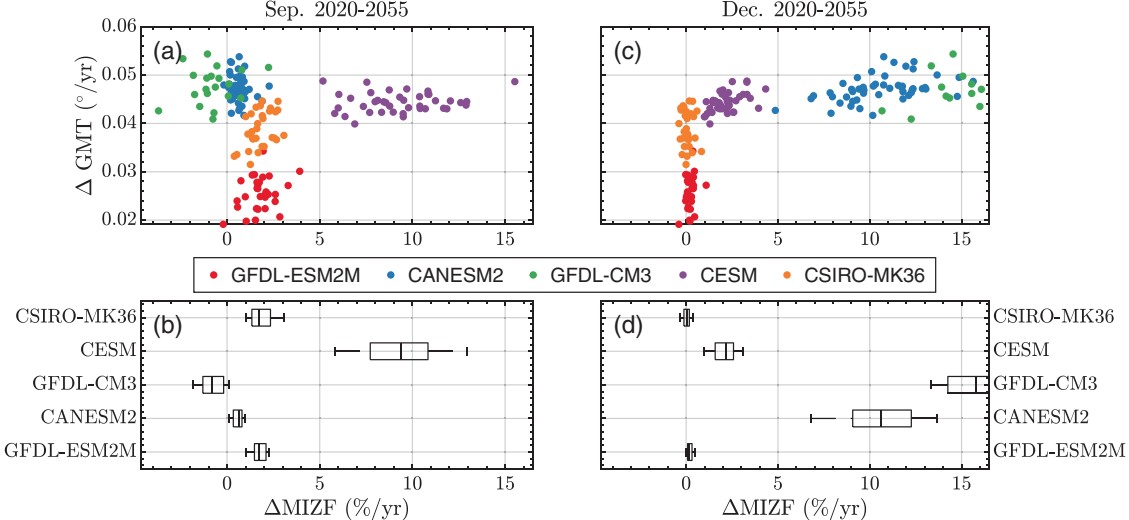

**Fig. 4 Future MIZF variability in the CLIVAR-LE. a** Scatter plot of October 2020–2055 trend in global mean temperature vs normalized trend in MIZF (expressed as a percentage per year). **b** Box plots of MIZF trends alone. **c**, **d** Same as (**a**, **b**) for December.

pathway at the time this manuscript was compiled. I choose September for three reasons: it is the month of the annual sea ice minimum, there is no significant correlation between historical ΔMIZF and GMT, and it is well-predicted, with 4/5 CLIVAR-LE members and the CMIP6-EME plausibly estimating historical September ΔMIZF. Because of the historical similarities between CLIVAR-LE members, a divergence in future projections of September ΔMIZF can help differentiate between models. December is chosen because it has the largest annual differences in predicted ΔMIZF between models.

On the basis of September and December SIA projections, the models fall into three general groups. Two (GFDL-ESM2M and CSIRO-MK36) show MIZF trends that are ≈2%/yr in September, but are indistinguishable from zero in December. Two others (GFDL-CM3 and CANESM2) have near-zero trends in September but large (>10%/yr) ΔMIZF in December. The final, CESM, has a positive MIZF trend in September but weaker positive trend (ΔMIZF ≈ 2%/yr) in December. There is a difference in ΔMIZF for models with similar predicted ΔGMT. For example, between CESM and GFDL-CM3, ensemble-mean values of ΔGMT vary by <8% (0.044°/yr vs 0.0472°/yr), and ensemble mean values of ΔMIZF vary by more than 600% (2.2%/yr vs 15.9%/yr).

## Discussion

Despite significant efforts to improve sea-ice models, GCMs underestimate the rate of September sea ice loss as a function of overall warming. Previous work examined whether this underestimation may be the result of internal variability in individual GCMs. Yet here, using five large ensembles and ensemble-mean output from CMIP6 over the period from 1979 to 2014, I show that the underestimation of September sea ice sensitivity persists and extends to the entire seasonal cycle. By using these ensembles to quantify both forced internal variability and inter-model variability, I show this underestimation cannot plausibly be explained as the result of either. As a consequence of the high correlation between SIA/SIE trends and GMT trends found across GCMs, it is challenging to understand whether improvements in modeled SIA/SIE over time are due to changes to sea-ice models themselves, or due to improvements in other components of GCMs that lead to better representation of GMT trends.

On the other hand, GCMs capably simulate changes in MIZF in most months of the year over 1979–2014, and changes to Arctic MIZF are less determined by changes to GMT. The result

of this weaker covariance, especially in September, suggests that changes in MIZF, for example between model versions or over time, can be used to evaluate whether sea-ice models and their modeled responses to warming are improving on their own.

Accounting for observational uncertainty and internal variability, all five members of the CLIVAR-LE produce plausible estimates of MIZF change over January-September. While they fail to do so from October-December, this bias (a) is during the annual period where MIZF is most determined by GMT change (Fig. 2), and (b) can be explained as a result of inter-model differences using the CMIP6-EME. Understanding how the representation of historical MIZF changes with changes to sea-ice models could be useful for evaluating improvements or declines in sea-ice model skill.

The results presented here are consistent with the general consensus that sea-ice models are not sufficiently sensitive to warming. However, the result that model internal variability cannot explain the sensitivity of September sea ice (nor most other months), to warming (Fig. 3, top row) differs from the analysis of refs. [8,17]. I suggest two main explanations for this difference. First, different data and methodology[17]: sought to solve the problem of assessing internal variability for small forced ensembles, using an ergodic assumption to convert control simulation variability to ensemble variability. Here, with a large number of members in each of the 5 + 1 ensembles, a more direct and simpler evaluation of internal variability is possible, one with a significantly tighter constraint on internal variability than is possible for smaller ensembles (see "Methods") and therefore a less permissive test of plausibility. In addition, in ref. [8], an average estimate of model internal variability was defined by pooling all CMIP6 models that contributed three or more simulations, and here I compute the model internal variability for each CMIP6 model on its own. Second, we examine slightly different fields. I find that trends in SIA alone are plausibly produced by the CMIP6-EME (Fig. 1e), however as shown here and identfied by refs. [1,8], the sensitivity of SIA to warming is not (Fig. 3). With a large spread in simulated mean state between model groups, I evaluate trends in dimensionless sea ice fields, though evaluating the trends of dimensional fields yields similar results (SI, Supporting Fig. S1). This choice is guided by the fact that sea ice feedbacks are highly state-dependent[33–36]. A loss of 1 million sq km of sea ice for a simulation with a mean state of 5 million sq km, versus a mean state of 10 million sq km may have the

same immediate geophysical impact but carries double the feedback effect.

MIZF is a quantity that suffers from a high level of observational uncertainty, which may increase the plausibility of modeled MIZF sensitivity to warming. Passive microwave estimates of sea ice concentration used to estimate MIZ coverage can provide significantly different estimates of local sea ice concentration for the same return, a particular challenge in summer when sea ice is covered in melt ponds[16]. Accordingly observational uncertainty is highest in these months (reaching 45% in July, for a trend of ±1.1%/yr over the historical period)—reducing the ability to differentiate plausible model estimates of $\rho$MIZF from implausible ones and potentially limiting the ability to differentiate model estimates from one another in the future. The wider error bounds for MIZF, however, do not affect the plausibility of modeled MIZF. The results presented here hold generally even if the uncertainty in observed MIZF is artificially reduced to be the same as for observed SIA (see "Methods", Supporting Fig. S2). In addition, projected increases in MIZF change over 2020–2055 for those models that expand in either September or December greatly exceed observational uncertainty in those months. Historical observational MIZF uncertainty is 37% (~1%/yr) in September and 4% (≈0.1 %/yr) in December—with "expanding" models growing at 10% or more per year, the signal of a growing MIZF will quickly emerge from the observational uncertainty especially in December. Improvements in retrieving the MIZF, for example through improved spectral unmixing and machine learning techniques[37] or new satellite methodologies[38] will permit cleaner discrimination of GCM output against observations.

Many of the sea-ice models used in modern GCMs share common parameterizations or development histories, and the interrelation of sea ice and ocean models could impact estimates of inter-model variability[39]. Developing a method to properly weight ensemble mean statistics based on climate model interdependence to form the CMIP-EME is not within the scope of this study, and I note that despite sharing the same sea ice and ocean model components, the two GFDL contributions to CLIVAR-LE diverge significantly in predictions of future MIZF.

Indeed, individual GCMs make highly differentiable predictions about future MIZF, which can be more readily disentangled from future warming than SIA or SIE. Examining projections of September and December MIZF under the RCP8.5 forcing scenario, the five CLIVAR-LE models can be loosely grouped into three categories: those predicting a large September MIZF expansion (CESM), those predicting a large December MIZF expansion (CANESM2,GFDL-CM3), and those predicting weak to no MIZF expansion over the next 35 years (GFDL-ESM2M, CSIRO-MK3.6). The magnitude of projected trends in the two "expanding" cases is much larger than historical values of MIZ: for example, CESM projects a 10%/yr increase in MIZF from 2020 to 2055 following the RCP8.5 pathway, but MIZF expanded by <1%/yr from 1979–2014. Because the projected signals of MIZF expansion are high (increases of 10% per year, compared to typical historical trends of 1–2% per year), and in September MIZF changes are not correlated with GMT changes, it may be possible to use MIZF forecasts in real time to assess and analyse the impact of changes to sea-ice model physics on Arctic sea ice, independent of whether models accurately represent warming.

Processes that affect the variability and expansion of the MIZ are present around the sea ice margin, and may not vary greatly in effect over time but simply move north as sea ice retreats as suggested by ref. [30]. This could explain why MIZF trends are less correlated to GMT trends in several months. MIZF trends are also poorly predicted by the models considered here during the period of sea ice freeze-up and expansion, from October to December. This may point to the parameterization of new ice

growth as an area of sea-ice model weakness, although the growth of new sea ice is highly tied to local temperature, and these months are also those with the highest correlation between MIZF trends and GMT trends. This period of the year is also that in which the effect of waves is most pronounced in Arctic sea ice[38], and none of the CLIVAR-LE models nor any models submitting to CMIP6 incorporate the coupled evolution of sea ice and waves, one of the most significant drivers of MIZ variability[40]. Progress is being made at incorporating wave and responsive floe size distribution models[29,41–43] in GCMs, and understanding how parameterizations of coupled wave, thermodynamic, and dynamic processes influence model estimates of MIZF will be a key area for future efforts.

## Methods

**CLIVAR large ensemble data.** I use five contributing members to the CLIVAR large ensemble that provide sea ice concentration fields to the EarthSystemGrid data portal (CSIRO-Mk3.6, GFDL-ESM2M, GFDL-CM3, CANESM2, and CESM1), which are listed in Table 1). These simulations are branched from pre-industrial spinups and output encompasses at least 1950–2100. Information about the initialization of and historical forcing of ensemble members is provided in ref. [20]. Historical forcing is applied until 2014, at which point model forcing follows the RCP8.5 pathway.

**CMIP6-EME data.** I also use eight available CMIP6 models with historical sea ice output available on the Earth System Frontier Grid as of 5/1/2020, that submitted 10 or more ensemble members, and are used to create the multi-model ensemble CMIP6-EME (listed in Table 1). As each individual modeling center submits a varying number of runs, the mean value of a statistic across all CMIP6 simulations is biased towards those models with more contributions. Thus CMIP6-EME is composed of "ensemble members" which are the ensemble mean value of the runs submitted by an individual model. The CMIP6-EME has a total of eight ensemble members. Internal variability expressed in the CMIP6-EME reflects inter-model variability. For example, while there are 11 contributions to CMIP6 from CESM2, one member of CMIP6-EME consists of the ensemble mean statistics from these 11 CESM2 simulations. Similarly, there are only two contributions from MPI-ESM-1-2-HAM: that model is not included as a CMIP6-EME member. No CMIP6 models contributed 10 ensemble members under the same RCP8.5 forcing scenario as the CLIVAR-LE, and therefore CMIP6-EME is not included in future projections.

**Global observations.** For observations of GMT, I use the un-smoothed NASA Global Land-Ocean Temperature Index, which spans 1880–2019. Observations of sea ice concentration are gridded daily data from three passive-microwave-derived estimates of sea ice concentration: NASATeam[44], Bootstrap[45], and OSISAF[46]. For comparison with model output, monthly maps are computed as the monthly average of the daily gridded product. Pole holes in the NASAteam and Bootstrap products are filled by taking the average concentration around the perimeter of the pole hole as in ref. [47]. The OSISAF-provided pole hole filling is used for that product.

**Calculation of global sea ice metrics.** Both observational and model products are available as gridded products. To avoid introducing additional bias through regridding, I compute all sea ice metrics on their original grid. I examine SIA and MIZ area (MIZA) rather than sea-ice extent to limit grid-related uncertainty[16].

SIA is calculated as the sum of sea ice concentration multiplied by grid cell area in all grid cells above 0°N. Marginal ice zone area (MIZA) is computed as the sum of sea ice concentration for all grid cells with <80% concentration. The MIZA fraction is then MIZF = MIZA/SIA. While SIA should not be sensitive to a changing grid, MIZA may be, as whether a given geographical region is above or below 80% sea ice concentration can depend on how the grid is drawn[16]. I do not consider this source of grid-related uncertainty here.

To evaluate linear trends, I perform a linear regression of the 36-year (1979–2014) time series against time and a constant term for each month (for sea ice products) or the annual average (for GMT). The trend $\Delta \mathrm{SIA}_9$, for example, is the slope of this linear fit for September SIA. Sensitivites are computed as the negative of the ratio of sea ice trends to annual GMT trends, for example:

$$\rho \mathrm{SIA}_9 = \frac{\Delta \mathrm{SIA}_9}{\Delta \mathrm{GMT}}. \tag{1}$$

Thus a negative value of $\rho X$ corresponds to a negative trend in $X$ for a positive $\Delta \mathrm{GMT}$.

**Uncertainty quantification.** The uncertainty associated with internal variability is quantified for each model using the unbiased sample standard deviation. Bessel's correction is applied to the sample variance, and Cochran's theorem corrects for the bias in the standard deviation. For an uncorrected sample standard deviation $\sigma_0$

over $N$ ensemble members, this ultimately corrects $\sigma$ upwards as:

$$\sigma = \sigma_0 \sqrt{\frac{N}{2}} \frac{\Gamma\left(\frac{N-1}{2}\right)}{\Gamma(N/2)}, \qquad (2)$$

where $\Gamma$ is the gamma function. For $N = 3$, as used in the pooled uncertainty metric by[8], this corrects the sample standard deviation upwards by 38%. For $N = 8$, as used to compile the CMIP6-EME ensemble, the correction is 11%. For $N = 20$, the smallest size of a CLIVAR-LE ensemble set, the correction is 4.0%.

For evaluating the plausibility of LE trends or statistics relative to observations, color coding in Figs. 1 and 3 are results of a two-sided (Welch's) $t$-test, where the null hypothesis is that the model and observational distributions are drawn from a normal distribution with the same mean but not necessarily the same variance, evaluated at the 95% confidence interval. The unequal variance test is chosen as model variance (related to internal variability) and observational variance (related to observational uncertainty) are not expected to be the same. Red values reject the null hypothesis at the 5% significance level. Whisker plots extend the model data three (unbiased) standard deviations from the mean, and dashed lines extend the observational estimate one (unbiased) standard deviation from the mean.

For visual comparison, assume the null hypothesis that ensemble statistics and observational statistics are drawn from normal distributions with different variances but the same mean. Then the likelihood that whiskers in Figs. 1 and 4 do not overlap is bounded above by $\exp(-6)/4 \approx .06\%$. The likelihood that box whiskers and dashed lines in Fig. 3 do not overlap is bounded above by $\exp(-4)/4 \approx 0.5\%$.

**Testing plausibility with artificially reduced observational uncertainty.** To test whether the increased plausibility of $\rho$MIZF estimates is due only to higher observational uncertainty, I create a synthetic dataset of $\rho$MIZF composed of the observational $\rho$MIZF mean and two-time series spaced at $\pm 2$ standard deviations from the mean, but where those standard deviations are computed instead as the observational uncertainty for $\rho$SIA. I reproduce Fig. 3 in the Supplementary Information (Supporting Fig. S2) using this artificially reduced uncertainty. The plausibility of model estimates of $\rho$MIZF are qualitatively similar compared to using the true (higher) observational uncertainty: the CMIP-EME is plausible in September, though 4/5 CLIVAR-LE models are not.

## Data availability

CLIVAR-LE data used in the model study is available via the NCAR Climate Data Gateway at https://www.earthsystemgrid.org/dataset/ucar.cgd.ccsm4.CLIVAR_LE.html. CMIP6 data used in this study are available at the Earth System Federated Grid at https://esgf-node.llnl.gov/projects/cmip6/ and were downloaded using code developed for the Sea-Ice Model Intercomparison Project[8,48] by Jakob Doerr.

## Code availability

Marginal Ice Zone Fraction Benchmarks Sea Ice and Climate Model Skill. https://doi.org/10.5281/zenodo.4321849 and Code[49] used to produce figures in this manuscript is available at https://github.com/chhorvat/MIZ-SIE/. Python code used to download CMIP6 data is provided by Jakob Doerr at https://github.com/jakobdoerr/SIMIP_2020 under an MIT license.

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

## Acknowledgements

C.H. thanks Jakob Doerr and Lettie Roach for their assistance in accessing and downloading CMIP6 data, Karen McKinnon for assistance in interpreting ensemble statistics, and Dirk Notz and Dirk Olonscheck for advice on presenting sea ice variables and trends. C.H. thanks the National Institute of Water and Atmospheric Research in Wellington, New Zealand, for their hospitality during this work. I acknowledge the World Climate Research Programme, which, through its Working Group on Coupled Modelling, coordinated and promoted CMIP6. I thank the climate modeling groups for producing and making available their model output, the Earth System Grid Federation (ESGF) for archiving the data and providing access, and the multiple funding agencies that support CMIP6 and ESGF. C.H. was supported by NASA grant 80NSSC20K0959.

## Author contributions

C.H. conceived the study, analyzed the data, and wrote the manuscript.

## Competing interests

The author declares no competing interests.
