## [Peer Review File · Nature Communications]

REVIEWER COMMENTS

Reviewer #1 (Remarks to the Author):

Review of the paper

'Global Climate Models Divided by their Representation of the Marginal Ice Zone'

C. Horvat

submitted to Nature Communications (NCOMMS-20-27272-T)

Over the last decades, one of the most pronounced imprints of global climate change is the rapid decrease of Arctic sea-ice area (SIA), which has clearly emerged from the large year-to-year variability. Therefore, the examination of climate model performance with respect to the observed changes in Arctic sea-ice area is very important.

When evaluating Arctic sea-ice area in climate models it is not enough to analyze the mean sea-ice fields and their trends, because so far, the simulations with Arctic sea ice retreat as fast as observations are found to have considerably positive biases in global warming (Rosenblum and Eisenman, 2017).

Even with the new CMIP6 generation of climate models, this situation only slightly improves. Although the CMIP6 multi-model ensemble mean provides a more realistic estimate of the sensitivity of September Arctic sea-ice area to a given amount of global warming, this sensitivity is generally underestimated (Notz et al, 2020). So far it is not clear, whether the slight improvements are caused by a change in the forcing or by improvements of model physics.

In extension to the earlier work of many authors analyzing the sensitivity of September Arctic sea-ice area to a given amount of global warming, the author showed that the latest generation of climate models do not plausibly estimate the sensitivity of sea ice area to global warming not only in September, but in almost any month.

In search for an alternative metric for the evaluation of climate model performance with respect to Arctic sea ice the author identified a new benchmark metric, the marginal ice zone fraction (MIZF). Since the MIZF is less correlated to warming biases and shows plausible estimates of its sensitivity to global warming from January to September for the analyzed climate model ensembles, this metric has the potential to relate bias reductions in MIZF to improvements in model physics.

In this way, the manuscript could deliver a valuable contribution to the development of suitable and robust metrics for evaluating the performance of climate models.

Nonetheless, I have several comments and suggestions to the manuscript regarding more careful interpretation and discussion of the results and subsequent conclusions which would require major revision.

The authors claim in the abstract, that model assessments now suggest trend biases may be explained as a result of model internal variability. But reading e.g. one of latest studies by Notz et al. (2020) I do not agree with that. Notz et al. (2020) wrote that the models show "general underestimation of the sensitivity of September sea-ice area to a given amount of global warming; and most models' failure to simulate at the same time a plausible evolution of sea-ice area and of GMST". In the same direction: On page 3 and in the discussion the authors claim that their results concerning the sensitivity of SIA to warming differs from earlier studies (e.g. Olonscheck and Notz (2017); Notz et al. (2020)). To agree with these statements given by the author more arguments and explanation about the different approaches in the mentioned earlier studies is required.

Furthermore, more information about the CMIP-EME ensemble is needed (which models with how many ensemble members contribute to CMIP-EME). This missing information do not allow to reproduce the results of this study. To ensure fully reproducibility of the results would require the version number (and doi, if available) of the used CMIP6 and CLIVAR-LE data.

Statistical tests for significance of correlations and also uncertainty quantification have been applied, but more explanation should be given why the specific t-test (where the null hypothesis is that the model and observational distributions are drawn from a normal distribution with the same mean but not necessarily the same variance) has been applied. Furthermore the distribution of the points in Fig. 3c suggests, that calculation of linear correlation is not appropriate (see below comment 9).

Further comments:

1) page 1: "... wide range of complexities (Gregory et al. 2002; Notz and Stroeve 2016)." There are much more studies on that issue, also from the last years, more references have to be given.

2) page 1: " MIZ location..."
atmosphere-ice-ocean interaction processes are crucial as well.

3) page 1: "..observed forced response"
The author has to be more precise. He showed the sensitivity of a change in SIA or MIZF to change in global warming, which is a particular response of the climate system to changes in external forcing!

4) page 2:
More information about the CMIP-EME ensemble is needed.

5) page 3: "The scatter of (Notz and Stroeve 2016)".
Wrong citation, in Notz and Stroeve (2016) the relation between SIA and cumulative CO2 emissions was shown.

6) page 3 left column: Fig.2 instead of Fig.3 shows the explained variance.

7) page 3 paragraph "The preceding..." Interpretation of and conclusions from Fig. 2 have to be added.

8)Page 3: "sea ice sensitivity" should be explained here in a few words.

9) page 4/5: Fig 4. and related text:
In Fig.4 a color code for the different models has to be added. The distribution of points in fig. 4c suggests that there is no linear correlation (no Gaussian distribution of the residuals), therefore it makes no sense to claim, that there is a significant correlation!

10) page 5 Discussion: "Because sea ice and..."
In my understanding, both biases, (biases in the response of global mean temperature to forcing and biases in sea ice models physics) contribute to the lack of skill in sea ice models. Please explain your conclusion in more detail.

"...estimates of MIZ fraction"
Since plausible estimates of MIZ fraction change have been estimated over the period from January-September, but not for October-December, the statement "MIZ fraction is therefore a useful benchmark for sea ice model and GCM skill" needs more justification.

Reviewer #2 (Remarks to the Author):

Summary

While this manuscript has interesting and potentially novel metrics, it's a bit difficult to see the forest for the trees in its current format. If the take-home message of the manuscript is given by the title, i.e., the future behaviour of the MIZF in the models, then much of the manuscript is overly detailed in my opinion. However, if the author would like to present a more general overview of the representation of the MIZF in the models, the manuscript needs a different title. Combined with a number of editorial mistakes, it is somewhat difficult to read currently.

Major concerns:

General

As I mentioned above, the title to me is not particularly representative of the manuscript. The division of 5 models into three classes is in future projections and most of the manuscript is focused on the historical period where, as I understand it, there is not a clear division in the representation of the MIZF. On this note, the part of the manuscript where the three classes are described (the paragraph beginning with "All five models can be clearly grouped") could use more physical motivation. Why should the models have different behaviours in September and December? Do the models that behave the same way share components of their sea ice models, or perhaps their ocean models? Additionally, dividing five models into three classes does not seem to be a strong or clear relationship. How does one determine which of these three behaviours represents a skillful sea ice model if none of the models exhibit any dependence on global mean temperature currently?

Throughout, I find the claim of forcing biases being to blame for the incorrect relationship between global mean surface temperature and sea ice area as compared to sensitivity biases to be confusing. I'm not certain I understand this conclusion, and either it requires further analysis on the part of the author or more explicit explanation with citations to existing literature.

Abstract:

Related to point 2 above, how does an anti-correlation between sea ice area trends and global warming trends suggest forcing biases are to blame?

Section 1

Paragraph 4: The greater observational uncertainty in Δ MIZF trends than in Δ SIA seems like it may be a design flaw in this study. This is addressed partially in the Discussions Paragraph 5 but mostly in terms of differentiating a trend in MIZF from noise, not in determining the plausibility of model agreement over the historical period.

Section 2

"CMIP6 biases might be explained by the consequence of inter-model variability" -- or is it an incorrect sensitivity of sea ice to warming?

What physical reason would explain the lack of correlation between MIZF and warming?

What happens in the autumn for the models to not simulate a plausible MIZF? These months also show the highest amount of explained variance of the MIZF with global mean temperature, more like SIA. Is the lack of plausibility in SIA and MIZF connected somehow?

Discussion Section

Last paragraph:

Is the lack of surface waves implying that the models are getting the MIZF right for the wrong reason? If true, it undermines quite a bit of the manuscript.

Minor concerns:

Abstract:

Cleanly divide GCMs into which three classes? Using a colon here makes it look like you are going to state them.

Figure 1

Why use black for both one of the ensembles and the observations?

Why is there no uncertainty estimates on the CMIP6-EME

There is a point going off the edge of plots (c) and (d)

In the caption: dashed line -> shaded area?

Observation -> observations

Figure 2

In the caption: CMIP6->CMIP6-EME?

Figure 3

Sea ice sensitivity, rho SIA

You should put your ensemble members in the same order in each figure for ease of comparison.

Figure 4

Include a legend on this one as well

Intro

P1

The reference here (and elsewhere in the manuscript) to Notz et al. 2020 should be SIMIP Community 2020.

P2

Very long run-on sentence to end the paragraph makes it difficult to follow

P3

However marginal ice zone extent -> missing comma after however.

P5

You switch between model-to-model variance and inter-model variance and it would be easier to read if you chose one and stuck to it.

MIZ sensitivity -> MIZF sensitivity? This is in other places in the manuscript as well and I believe they should almost all be MIZF. Also MIZ fraction is in the sentence below and should just stick with MIZF.

Section: GCM estimates of September Sea Ice and MIZ change -- (MIZF change?)

P1

Which CMIP ensemble are you referring to here? CMIP-EME or all CMIP5(6?) models?

P2

Isn't percentage always unitless?

"Assuming sea ice changes consistent with a feedback response to warming" -> what does this mean?

With red boxes models -> with red boxes indicating models

And blue those -> and blue boxes indicating those models

Below I show as in... -> below I show, as in..

P3

Mix of em dash and comma in the sentence beginning "All three observational estimates..."

Section: Seasonal GCM estimates of SIA and MIZ sensitivities

P1

Is model internal variability something different than intra-model variability?

P2

Positive sensitivity corresponds to a declining field - confusing definition and it seems like Figure 3 is plotting negative values so wouldn't that mean negative sensitivity corresponds to a declining

field?

Use an em dash for the sentence "Red boxes plot..." instead of a comma

Negative sensitivity -> positive sensitivity? Or is it largest magnitude sensitivity?

When examining different fields - which fields?

P3

MIZ sensitivity is rhoMIZF so shouldn't it be MIZF sensitivity

Start a new paragraph with "Despite producing..." because the topic has changed

How does inter-model spread explain MIZF sensitivity?

Section: Future projections of MIZ variability (MIZF variability?)

P1

Section: Discussion

P2

This bias (a) is -> this bias is (a).

P4

...and those predict... -> and those predicting

Data and Methods

Which members are in the CMIP6-EME?

Similarly there are only two.... -> This is obvious and doesn't need to be explicitly stated

Is the reason for a lack of CMIP6 future forcing scenarios of RCP8.5 because of the new projection pathways or because they don't provide enough future members under any scenario?

Sea ice observations of sea ice concentration - redundant

Switch between passive and active voice here

SIA is calculated as..-> multiplied the area of the cell, correct?

Notz (2014) -> (Notz, 2014)

All supplemental figures are lacking captions

Reviewer #3 (Remarks to the Author):

Review of "Global Climate Models Divided by their Representation of the Marginal Ice Zone"

Using model ensembles from CMIP6 and the CLIVAR-LE, this manuscript attempts to further understand the high sensitivity of Arctic sea ice area to warming in climate models. Findings show that for nearly all months, none of the models reasonably estimate the sea ice area sensitivity to warming, and that this is not a result of internal variability or inter-model differences. To combat this, a new metric, the MIZF (marginal ice zone fraction), is developed that is less sensitive to warming biases, and that can ultimately be used to compare sea ice model skill. The manuscript is sound, and the conclusions are appropriately supported by the figures/results. Overall this work represents a valuable contribution to the sea ice community (and the GCM community), and I recommend it for publication after the following minor comments/revisions are addressed.

Minor comments:

1. In the introduction you say that the CMIP6-EME has 8 members but in the methods you say that it has 10 members. Later on in that part of the methods you explain that you at least left 1 model (MPI-HAM) out of the EME because it didn't have enough members. Was this the case for the second model that was left out? I think it would help if you add another sentence in the methods clarifying that the total number of members in the EME is 8.

2. Which 8 CMIP6 models are used in the EME? I am also curious about this because some CMIP6 models use the same sea ice and ocean models, and I wondered if you consider such models as independent or not when computing statistics for the EME.

3. There are a number of places in the text where parenthetical figure references could be used in lieu of lengthy figure explanations. This would also help shorten the text and/or free up space for further discussion. For example, in the section "Seasonal GCM estimates of SIA and MIZ sensitivities", the sentence that starts with "In Fig. 3..." could be removed and the next sentence describing the correlation results could parenthetically cite Fig. 3 instead. In some cases, the figure descriptions are also better suited to the captions. This is particularly true for parts of the description of Fig. 3 such as "Whiskers extend the boxes +/-3 standard deviations from the mean in each month, and are plotted against the observed annual cycle of SIA sensitivity (black), with observational uncertainty shown as +/-1 standard deviation of the observational mean."

4. Units of Kelvin do not have a degree symbol

Minor figure suggestions:

1. Figure 1a: The black dashed line is somewhat difficult to distinguish in this panel. I wonder if a dotted line would be better here although it could very well make things worse. I hesitate to suggest adding another color to the plot but that might also be an option. If you already tried all of that feel free to ignore this.

2. Figure 4 is missing a legend for the scatterplot color associated with each model

Line-by-line (but without line numbers so this is going to be a fun exercise in figuring out whether or not I give good directions. Good luck out there!):

1. Beginning of paragraph 4 of the intro: "respnse" should be "response"

2. Final paragraph of intro beginning with "Across this set of 5+1 ensembles...": You can probably just say 6 here even if there are two different types of ensembles. Also please decapitalize the "s" in GCMS

3. Figure 1 caption: "observation" should be pluralized in "Black lines are observation." Same with "Crosses are observation."

4. Paragraph 2 of "GCM estimates of September Sea Ice and MIZ Change": I suggest rewriting the 3rd sentence as "Box plots of Δ SIA for each model are given in Fig. 1(e). Red boxes indicate models that produce implausible (see methods) estimates of Δ SIA given the observations (black box labeled OBS, Fig. 1e), and blue those that produce plausible estimates." Or even better, I suggest getting rid of most of this and making all of the references to Fig. 1 parenthetical as I mentioned in the minor comments section.

5. And the next sentence "While individual CLIVAR-LE models do not individually produce...": "individually" is redundant and can be omitted

6. Paragraph 3 of "GCM estimates of September Sea Ice and MIZ Change": I suggest rewriting the beginning of the first sentence as "the same analysis for marginal ice zone fraction (MIZF; Fig. 1b,d,f) reveals.."

7. In the next sentence, "satellite" should be "satellites"

8. Figure 3 caption: 1. Second sentence: I suggest a minor rephrasing to "Boxes cover the interquartile range and horizontal whiskers the ensemble median." 2. Later on there should be a "the" between "is" and "mean" in this sentence: "Black solid line is mean from observations."

9. Discussion, 2nd paragraph, first sentence: you can get rid of "period from" in this

10. Discussion, 3rd paragraph, sentence that starts with "Evaluating the trends of dimensionful fields..." I have never heard the term "dimensionful" although the internet tells me it's a word haha. I still think dimensional is a better word here, but it's your call

11. Discussion, 3rd paragraph, last sentence: there is an extraneous "versus" in this sentence after "10 million sq km"

12. Discussion, 4th paragraph, 1st sentence, "predict" should be "predicting" to make it consistent with the earlier parts of the sentence

13. Last sentence of the 1st paragraph of the CMIP6 data section in the methods: "Internal variability expressed in the CMIP6-LE reflects model-to-model variability." Should that be the CMIP6-EME?

14. Last sentence of the 2nd paragraph of the "uncertainty quantification" section of the methods: "extent" should be "extend"

Thanks to the three reviewers for the efforts in reviewing this work. I believe their efforts have made the manuscript significantly more coherent and clearer. All reviewers suggested more information be provided on the models analyzed in the study, and I added a table identifying all models used, along with information regarding their sea ice and ocean components, following CMIP6 data citation guidelines. Further discussion of sea ice model inter-dependencies and the processes that determine MIZ variability are included in the text as suggested. Additionally, following the advice of Reviewer 2, the title of the manuscript is now altered to more accurately reflect the main arguments in this study.

All reviewer comments are reproduced below in blue. Responses and line numbers of changes are in black text. Where appropriate, manuscript changes are provided in indented quotes. I hope these revisions render this work suitable for publication in *Nature Communications*, and look forward to further correspondence.

Christopher Horvat

Reviewer 1:

Over the last decades, one of the most pronounced imprints of global climate change is the rapid decrease of Arctic sea-ice area (SIA), which has clearly emerged from the large year-to-year variability. Therefore, the examination of climate model performance with respect to the observed changes in Arctic sea-ice area is very important.

When evaluating Arctic sea-ice area in climate models it is not enough to analyze the mean sea-ice fields and their trends, because so far, the simulations with Arctic sea ice retreat as fast as observations are found to have considerably positive biases in global warming (Rosenblum and Eisenman, 2017).

Even with the new CMIP6 generation of climate models, this situation only slightly improves. Although the CMIP6 multi-model ensemble mean provides a more realistic estimate of the sensitivity of September Arctic sea-ice area to a given amount of global warming, this sensitivity is generally underestimated (Notz et al, 2020). So far it is not clear, whether the slight improvements are caused by a change in the forcing or by improvements of model physics.

In extension to the earlier work of many authors analyzing the sensitivity of September Arctic sea-ice area to a given amount of global warming, the author showed that the latest generation of climate models do not plausibly estimate the sensitivity of sea ice area to global warming not only in September, but in almost any month.

In search for an alternative metric for the evaluation of climate model performance with respect to Arctic sea ice the author identified a new benchmark metric, the marginal ice zone fraction (MIZF). Since the MIZF is less correlated to warming biases and shows plausible estimates of its sensitivity to global warming from January to September for the analyzed climate model ensembles, this metric has the potential to relate bias reductions in MIZF to improvements in model physics.

In this way, the manuscript could deliver a valuable contribution to the development of suitable and robust metrics for evaluating the performance of climate models.

Thank you for this helpful summary and contextualization of the results. Please see below where I add detailed responses to individual comments:

Nonetheless, I have several comments and suggestions to the manuscript regarding more careful interpretation and discussion of the results and subsequent conclusions which would require major revision.

The authors claim in the abstract, that model assessments now suggest trend biases may be explained as a result of model internal variability. But reading e.g. one of latest studies by Notz et al. (2020) I do not agree with that. Notz et al. (2020) wrote that the models show “general underestimation of the sensitivity of September sea-ice area to a given amount of global warming; and most models’ failure to simulate at the same time a plausible evolution of sea-ice area and of GMST”.

In the same direction: On page 3 and in the discussion the authors claim that their results concerning the sensitivity of SIA to warming differs from earlier studies (e.g. Olonscheck and Notz (2017); Notz et al. (2020)). To agree with these statements given by the author more arguments and explanation about the different approaches in the mentioned earlier studies is required.

Thank you for raising these points - more careful definitions are required within the text of this manuscript. There are two: first, the “general underestimation” of sensitivities in CMIP models, and second, whether that underestimation could be explained as the result of internal variability. The distinction was not clear enough in the initial submission. In the abstract, I now write:

Global climate models (GCMs) consistently underestimate the loss of September Arctic sea-ice area, and exhibit a linear correlation between modeled sea-ice area losses and the magnitude of modeled global mean temperature increases. This correlation makes it challenging to gauge whether changes in the representation of Arctic sea ice in GCMs derive from biases in sea ice model physics or from biases in modeled global mean temperature, driven by other GCM components. As climate models and sea-ice models have increased in complexity, multi-model assessments have demonstrated that September sea-ice area biases have declined, and are potentially explained as the consequence of model internal variability.

And when discussing the underestimation of sensitivities (page 4, line 190):

In all months, and across all of the ensembles, observed values of ρ SIA are more negative than model estimates. This includes an underestimation of September sea ice sensitivity, as previously observed by many (e.g., *Stroeve et al., 2007; Rosenblum and Eisenman, 2017; SIMIP Community, 2020*). However, controlling for internal variability (in the CLIVAR-LE) or inter-model variability (in the CMIP-EME), no models make plausible estimations of September ρ SIA, or in more than 3 months of the year (blue boxes). This differs from the results of *Olonscheck and Notz (2017); SIMIP Community (2020)*, who found that estimates of September sea ice sensitivity in CMIP5/6 class models were plausible when accounting for internal variability - although in those studies internal variability was quantified using different methodologies to account for having fewer ensemble members (see Discussion, below).

And in the discussion section, I make this distinction more explicit (page 6, line 313):

The results presented here are consistent with the general consensus that sea-ice models are not sufficiently sensitive to warming. However, the result that model internal variability cannot explain the sensitivity of September sea ice (nor most other months), to warming (Fig. 3, top row) differs from the analysis of *Olonscheck and Notz (2017)*; *SIMIP Community (2020)*. I suggest two main explanations for this difference. First, different data and methodology: *Olonscheck and Notz (2017)* sought to solve the problem of assessing internal variability for small forced ensembles, using an ergodic assumption to convert control simulation variability to ensemble variability. Here, with a large number of members in each of the 5+1 ensembles, a more direct and simpler evaluation of internal variability is possible, one with a significantly tighter constraint on internal variability than is possible for smaller ensembles (see Methods) and therefore a less permissive test of plausibility. Additionally, in *SIMIP Community (2020)*, an average estimate of model internal variability was defined by pooling all CMIP6 models that contributed 3 or more simulations, and here I compute the model internal variability for each CMIP6 model on its own.

Furthermore, more information about the CMIP-EME ensemble is needed (which models with how many ensemble members contribute to CMIP-EME). This missing information do not allow to reproduce the results of this study. To ensure fully reproducibility of the results would require the version number (and doi, if available) of the used CMIP6 and CLIVAR-LE data.

Indeed this is a good point and should have been included. A table in the methods section includes details on all models used in this study, the number of ensemble members, and the sea ice and ocean components of each model, along with DOI information to access the model output. Please see this in the methods section.

Statistical tests for significance of correlations and also uncertainty quantification have been applied, but more explanation should be given why the specific t-test (where the null hypothesis is that the model and observational distributions are drawn from a normal distribution with the same mean but not necessarily the same variance) has been applied.

I explain why this specific t-test is chosen in the text: (pg 8,line 530),

The unequal variance test is chosen as model variance (related to internal variability) and observational variance (related to observational uncertainty) are not expected to be the same.

Furthermore the distribution of the points in Fig. 3c suggests, that calculation of linear correlation is not appropriate (see below comment 9).

This is true, I appreciate the correction. The sentence about linear correlation has been removed from the text.

(1) page 1: "... wide range of complexities (Gregory et al. 2002; Notz and Stroeve 2016)." There are much more studies on that issue, also from the last years, more references have to be given.

Of course! I have expanded this citation as follows (page 1, line 18):

The use of SIE/SIA to evaluate sea ice model improvements is confounded by the parametric relationship between modeled SIA and carbon emissions; and by extension global mean temperature (GMT), one that holds across sea ice and climate models of a wide range of complexities (*Gregory et al.*, 2002; *Armour et al.*, 2011; *Winton*, 2011; *Mahlstein and Knutti*, 2012; *Notz and Stroeve*, 2016; *SIMIP Community*, 2020, and references within).

2) page 1: " MIZ location..." atmosphere-ice-ocean interaction processes are crucial as well.

I added this to the text along with a citation (page 2, line 57),

MIZ location and variability is set through the coupled interaction of oceanic, atmospheric, thermodynamic and sea ice dynamical processes that have been the focus of many recent modeling efforts (*Williams et al.*, 2013; *Horvat and Tziperman*, 2015; *Manucharyan and Thompson*, 2017; *Wenta and Herman*, 2018; *Roach et al.*, 2018).

3) page 1: "...observed forced response" The author has to be more precise. He showed the sensitivity of a change in SIA or MIZF to change in global warming, which is a particular response of the climate system to changes in external forcing!

Indeed: the use of "forced response" was loose. This sentence has been restructured in the abstract:

I identify a new benchmark metric, the marginal ice zone fraction (MIZF), less correlated to warming biases and with a response to warming plausibly simulated by modern climate models from January-September.

4) page 2: More information about the CMIP-EME ensemble is needed.

See the above comment and table where the CMIP-EME model members are outlined.

5) page 3: "The scatter of (Notz and Stroeve 2016)". Wrong citation, in Notz and Stroeve (2016) the relation between SIA and cumulative CO2 emissions was shown.

Indeed, thank you for catching this. The intended reference (Notz and Stroeve 2018) along with more illustrative references are included now (page 2, line 130),

The scatter of ΔGMT vs. ΔSIA in Fig. 1(c) illustrates their known parametric relationship (*Winton*, 2011; *Mahlstein and Knutti*, 2012; *Notz and Stroeve*, 2018).

6) page 3 left column: Fig.2 instead of Fig.3 shows the explained variance.

Fixed, thanks!

7) page 3 paragraph "The preceding..." Interpretation of and conclusions from Fig. 2 have to be added.

The description of Fig.2 has been moved slightly earlier in the manuscript and merged with a discussion of the explained variance in September (page 2, line 155),

In Fig. 2 I plot coefficients of determination (Pearson's r^2) between annual ΔGMT and

monthly Δ SIA (red) and Δ MIZF (blue) trends, with significant correlations filled circles. SIA trends are significantly correlated with GMT trends in all months, whereas MIZF has no significant correlation ($p > 0.01$) with GMT in four: February, May, August, and September. The coefficient of determination of SIA exceeds that of MIZF in all months. In September, with a bivariate correlation coefficient of $r = 0.17$, only 3% of the variability in Δ MIZF across the simulations can be explained by Δ GMT variability.

8) Page 3: "sea ice sensitivity" should be explained here in a few words.

I now do (pg 4, line 176),

Fig. 3(top row) shows box plots the seasonal cycle of sea ice sensitivity, which is defined as the percent change in sea ice extent per unit of warming (ρ SIA, units %/K), for each of the 5+1 ensembles.

9) page 4/5: Fig 4. and related text: In Fig.4 a color code for the different models has to be added.

A legend is now added to Figure 4 as suggested.

The distribution of points in fig. 4c suggests that there is no linear correlation (no Gaussian distribution of the residuals), therefore it makes no sense to claim, that there is a significant correlation!

This is a good point to make. The mention of linear correlation across the models has been removed from the text.

10) page 5 Discussion: "Because sea ice and..." In my understanding, both biases, (biases in the response of global mean temperature to forcing and biases in sea ice models physics) contribute to the lack of skill in sea ice models. Please explain your conclusion in more detail.

Indeed, this sentence has been re-written to make this point more clearly (page 6, line 285):

As a consequence of the high correlation between SIA/SIE trends and GMT trends found across GCMs, it is challenging to understand whether improvements in modeled SIA/SIE over time are due to changes to sea-ice models themselves, or due to improvements in other components of GCMs that lead to better representation of GMT trends.

"...estimates of MIZ fraction ..." Since plausible estimates of MIZ fraction change have been estimated over the period from January-September, but not for October-December, the statement "MIZ fraction is therefore a useful benchmark for sea ice model and GCM skill" needs more justification.

This sentence is amended and expanded to be more clear about how this is a useful benchmark (page 6, line 292),

On the other hand, GCMs capably simulate changes in the MIZ fraction in most months of the year over 1979-2014 (MIZF). Changes to Arctic marginal ice zone fraction are less determined by changes to global mean temperature. The result of this weaker covariance, especially in September, suggests that changes in MIZF, for example between model versions or over time, can be used to evaluate whether sea ice models and their

modeled responses to warming are improving on their own.

Reviewer 2:

While this manuscript has interesting and potentially novel metrics, it's a bit difficult to see the forest for the trees in its current format. If the take-home message of the manuscript is given by the title, i.e., the future behaviour of the MIZF in the models, then much of the manuscript is overly detailed in my opinion. However, if the author would like to present a more general overview of the representation of the MIZF in the models, the manuscript needs a different title. Combined with a number of editorial mistakes, it is somewhat difficult to read currently.

Thank you for your efforts to understand and assess this manuscript. I hope in revising the manuscript the results have become clearer to interpret and more in line with the manuscript itself.

The reviewer's point about titling is well-taken, as this manuscript is not an assessment of MIZ projections but a comparison of MIZF and SIA. The title has been changed to: "Marginal Ice Zone Fraction, not Sea Ice Area, as a Benchmark for Assessing Sea Ice and Global Climate Model Skill".

Major concerns:

General As I mentioned above, the title to me is not particularly representative of the manuscript. The division of 5 models into three classes is in future projections and most of the manuscript is focused on the historical period where, as I understand it, there is not a clear division in the representation of the MIZF.

Please see above, as the title of the manuscript is now changed - though if the reviewer has suggestions for amending it I am also happy to take that into account.

On this note, the part of the manuscript where the three classes are described (the paragraph beginning with "All five models can be clearly grouped") could use more physical motivation. Why should the models have different behaviours in September and December? Do the models that behave the same way share components of their sea ice models, or perhaps their ocean models?

I explain in the text why September and December are chosen now (page 4, line 248):

I choose September for three reasons: it is the month of the annual sea ice minimum, there is no significant correlation between historical Δ MIZF and GMT, and it is well-predicted, with 4/5 CLIVAR-LE members and the CMIP6-EME plausibly estimate historical September ρ MIZF. Because of the historical similarities between CLIVAR-LE members, a divergence in future projections of September ρ MIZF can help differentiate between models. December is chosen because it has the largest annual differences in predicted ρ MIZF between models.

In the discussion I discuss the question of model component interrelation (page 6, line 383):

Many of the sea ice models used in modern GCMs share common parameterizations or development histories, and the interrelation of sea ice and ocean models could impact estimates of model-to-model variability (*Knutti et al., 2017*). Yet developing a method to properly weight ensemble mean statistics based on climate model interdependence to form the CMIP-EME is not within the scope of this study, and I note that despite sharing the same sea ice and ocean model components, the two GFDL contributions to

CLIVAR-LE diverge significantly in predictions of future MIZF.

A table of the CLIVAR-LE and CMIP6 models, with their constituent sea ice and ocean models and other relevant features is now given in the text.

Additionally, dividing five models into three classes does not seem to be a strong or clear relationship. How does one determine which of these three behaviours represents a skillful sea ice model if none of the models exhibit any dependence on global mean temperature currently?

I now more explicitly describe the point about discriminating sea ice models to indicate how this classification could be used in the future (page 7, line 408):

Because the projected signals of MIZF expansion are high (increases of 10% per year, compared to typical historical trends of 1-2% per year), and in September MIZF changes are not correlated with GMT changes, it may be possible to use MIZF forecasts in real time to assess and analyse the impact of changes to sea-ice model physics on Arctic sea ice, independent of whether models accurately represent warming.

Throughout, I find the claim of forcing biases being to blame for the incorrect relationship between global mean surface temperature and sea ice area as compared to sensitivity biases to be confusing. I'm not certain I understand this conclusion, and either it require further analysis on the part of the author or more explicit explanation with citations to existing literature.

This point has clearly not been well-presented. I now re-wrote the initial introduction of this point in the abstract:

Global climate models (GCMs) consistently underestimate the loss of September Arctic sea-ice area, and exhibit a linear correlation between modeled sea-ice area losses and the magnitude of modeled global mean temperature increases. This correlation makes it challenging to gauge whether changes in the representation of Arctic sea ice in GCMs derive from biases in sea ice model physics or from biases in modeled global mean temperature, driven by other GCM components.

I changed the verbiage, and re-wrote a sentence in the introduction to not so clearly blame forcing biases (pg 1, line 32),

The underestimation of September SIA loss has improved across CMIP generations, but the link between SIA and GMT makes it challenging to judge whether improvements in modeled sea ice originate in improvements in sea-ice model physics or are due primarily to improvements in the external forcing of sea ice (*Olonscheck et al.*, 2019), like changes to aerosol forcing (*Rosenblum and Eisenman*, 2016; *SIMIP Community*, 2020).

Abstract: Related to point 2 above, how does an anti-correlation between sea ice area trends and global warming trends suggest forcing biases are to blame?

This is now more clearly spelled out in the abstract:

Global climate models (GCMs) consistently underestimate the loss of September Arctic

sea-ice area, and exhibit a linear correlation between modeled sea-ice area losses and the magnitude of modeled global mean temperature increases. This correlation makes it challenging to gauge whether changes in the representation of Arctic sea ice in GCMs derive from biases in sea ice model physics or from biases in modeled global mean temperature, driven by other GCM components.

Section 1 Paragraph 4: The greater observational uncertainty in deltaMIZF trends than in deltaSIA seems like it may be a design flaw in this study. This is addressed partially in the Discussions Paragraph 5 but mostly in terms of differentiating a trend in MIZF from noise, not in determining the plausibility of model agreement over the historical period.

Yes! To my chagrin, in the text I did not reference a paragraph in the Methods, and a figure in the Supporting Information, where this point was addressed explicitly using artificially reduced Δ MIZF uncertainties. This is now explicitly mentioned in the text (page 6, line 364):

The wider error bounds for MIZF, however, do not affect the plausibility of modeled MIZF. The results presented here hold generally even if the uncertainty in observed MIZF is artificially reduced to be the same as for observed SIA (see Methods, Fig. S2).

Section 2 "CMIP6 biases might be explained by the consequence of inter-model variability" – or is it an incorrect sensitivity of sea ice to warming?

Indeed, this sentence was not explanatory enough and has been expanded (page 3, line 165):

September MIZF trends are also not correlated with GMT trends, and no members of the CLIVAR-LE plausibly estimate the observed change in September SIA. Still, the bias in modeled September SIA could plausibly be explained as a consequence of inter-model variability in the CMIP6-EME, though this does not take into account the correlation between GMT and September SIA. Here I investigate this correlation, examining SIA and MIZF sensitivity to warming across the seasonal cycle,

What physical reason would explain the lack of correlation between MIZF and warming?

This is now discussed in the discussion section: (page 7, line 416)

Processes that affect the variability and expansion of the MIZ are present around the sea ice margin, and may not vary greatly in effect over time but simply move north as sea ice retreats as suggested by (Rolph *et al.*, 2020). This could explain why MIZF trends are less correlated to GMT trends in several months.

What happens in the autumn for the models to not simulate a plausible MIZF? These months also show the highest amount of explained variance of the MIZF with global mean temperature, more like SIA. Is the lack of plausibility in SIA and MIZF connected somehow?

These three questions are worth discussing, and I added a paragraph in the Discussion to address them, incorporating a later paragraph on wave physics. (page 7, line 416)

Processes that affect the variability and expansion of the MIZ are present around the sea

ice margin, and may not vary greatly in effect over time but simply move north as sea ice retreats as suggested by (Rolph *et al.*, 2020). This could explain why MIZF trends are less correlated to GMT trends in several months. MIZF trends are also poorly predicted by the models considered here during the period of sea ice freezeup and expansion, from October-December. This may point to the parameterization of new ice growth as an area of sea ice model weakness, although the growth of new sea ice is highly tied to local temperature, and these months are also those with the highest correlation between MIZF trends and GMT trends. This period of the year is also that in which the effect of waves is most pronounced in Arctic sea ice (Horvat *et al.*, 2020), and none of the CLIVAR-LE models nor any models submitting to CMIP6 incorporate the coupled evolution of sea ice and waves, one of the most significant drivers of MIZ variability is ocean surface waves (Squire, 2007). Progress is being made at incorporating wave and responsive floe size distribution models (Bennetts *et al.*, 2017; Roach *et al.*, 2018; Bateson *et al.*, 2020; Boutin *et al.*, 2020) in GCMs, and understanding how parameterizations of coupled wave, thermodynamic, and dynamic processes influence model estimates of MIZF will be a key area for future efforts.

Discussion Section Last paragraph: Is the lack of surface waves implying that the models are getting the MIZF right for the wrong reason? If true, it undermines quite a bit of the manuscript.

The inclusion of information about waves was to highlight future comparisons and should have been better justified. This paragraph was merged into a discussion of MIZF evolution (page 7, line 416):

Processes that affect the variability and expansion of the MIZ are present around the sea ice margin, and may not vary greatly in effect over time but simply move north as sea ice retreats as suggested by (Rolph *et al.*, 2020). This could explain why MIZF trends are less correlated to GMT trends in several months. MIZF trends are also poorly predicted by the models considered here during the period of sea ice freezeup and expansion, from October-December. This may point to the parameterization of new ice growth as an area of sea ice model weakness, although the growth of new sea ice is highly tied to local temperature, and these months are also those with the highest correlation between MIZF trends and GMT trends. This period of the year is also that in which the effect of waves is most pronounced in Arctic sea ice (Horvat *et al.*, 2020), and none of the CLIVAR-LE models nor any models submitting to CMIP6 incorporate the coupled evolution of sea ice and waves, one of the most significant drivers of MIZ variability is ocean surface waves (Squire, 2007). Progress is being made at incorporating wave and responsive floe size distribution models (Bennetts *et al.*, 2017; Roach *et al.*, 2018; Bateson *et al.*, 2020; Boutin *et al.*, 2020) in GCMs, and understanding how parameterizations of coupled wave, thermodynamic, and dynamic processes influence model estimates of MIZF will be a key area for future efforts.

Minor concerns:

Abstract:

Cleanly divide GCMs into which three classes? Using a colon here makes it look like you are going to state them.

I rewrote this sentence: (page 2, line 96):

As future projections of September and December MIZF differ radically between models, MIZF may also be a useful way to determine in real-time whether current models make skillful predictions of future Arctic sea ice variability.

Figure 1 Why use black for both one of the ensembles and the observations? Why is there no uncertainty estimates on the CMIP6-EME There is a point going off the edge of plots (c) and (d) In the caption: dashed line → shaded area? Observation → observations

The CMIP6-EME is now plotted in brown, with uncertainty shading for inter-model variance. Y limits have been adjusted slightly to keep all points within the plots, and the caption edits made as suggested. Please see the new Fig. 1.

Figure 2 In the caption: CMIP6→CMIP6-EME?

Fixed here and in the new caption on Fig 6.

Figure 3 Sea ice sensitivity, rho SIA. You should put your ensemble members in the same order in each figure for ease of comparison.

Done, thanks!

Figure 4 Include a legend on this one as well

Done!

Intro P1 The reference here (and elsewhere in the manuscript) to Notz et al. 2020 should be SIMIP Community 2020.

Thanks! The SIMIP manuscript was published as Notz (2020) at the time this manuscript was compiled.

P2 Very long run-on sentence to end the paragraph makes it difficult to follow

This sentence has been restructured: (page 1, line 41)

Instead, *Olonscheck and Notz (2017)* used year-to-year variance from a single model realization over a long pre-industrial simulation to represent model internal variability. The recent publication of large (20+ member) ensembles (*Kay et al., 2015; McKinnon and Deser, 2018; Deser et al., 2020*), and the contribution of multiple historical runs to the 6th CMIP from a number of modeling centers (see Table 1) now allows for a direct evaluation of model internal variability for several independent models, and over the same period that sea ice has declined.

P3 However marginal ice zone extent → missing comma after however.

Thanks!

P5 You switch between model-to-model variance and inter-model variance and it would be easier to read if you chose one and stuck to it.

Great point. This is now consistent throughout as inter-model variance.

MIZ sensitivity → MIZF sensitivity? This is in other places in the manuscript as well and I believe they should almost all be MIZF. Also MIZ fraction is in the sentence below and should just stick with MIZF.

Good point - this has been edited to be more consistent throughout as either MIZF or MIZ fraction (where appropriate)

Section: GCM estimates of September Sea Ice and MIZ change – (MIZF change?)

Thanks! Fixed.

P1 Which CMIP ensemble are you referring to here? CMIP-EME or all CMI5(6?) models?

This has been clarified (page 2, line 114):

Consistent with the model-mean behavior of the 5th and 6th CMIPs (*Stroeve et al.*, 2012; *SIMIP Community*, 2020), while there is broad disagreement in the magnitude of SIA among individual models, the CMIP6-EME mean generally tracks observed September SIA.

P2 Isn't percentage always unitless? "Assuming sea ice changes consistent with a feedback response to warming" → what does this mean? With red boxes models → with red boxes indicating models And blue those → and blue boxes indicating those models Below I show as in... → below I show, as in..

This paragraph has been cleaned up and minor edits made as suggested.

Because of the large difference in modeled SIA and MIZF mean states, trends are expressed as a percentage change per year for these fields (this choice is examined in detail in the Discussion).

P3 Mix of em dash and comma in the sentence beginning "All three observational estimates..."

I removed the punctuation ambiguity here (page 2, line 143):

All three observational estimates show positive Δ MIZF from 1979-2014. The CMIP-EME and 4 of 5 members of the CLIVAR-LE produce plausible estimates of September Δ MIZF over this period.

Section: Seasonal GCM estimates of SIA and MIZ sensitivities P1 Is model internal variability something different than intra-model variability?

This sentence has been rewritten (page 3, line 161):

The preceding analysis demonstrated that September MIZF trends alone are plausibly estimated when considering both internal variability (in 4/5 CLIVAR-LE models) and inter-model variability (computed via the CMIP6-EME).

P2 Positive sensitivity corresponds to a declining field - confusing definition and it seems like Figure 3 is plotting negative values so wouldn't that mean negative sensitivity corresponds to a declining field?

This was restructured (page32, line 169)

Here I investigate this correlation, examining SIA and MIZF sensitivity to warming across the seasonal cycle, where the sea ice sensitivity (or MIZF sensitivity) ρ SIA is defined as the percent change in sea-ice extent per degree change in GMT (ρ SIA, units %/K). A declining SIA with increasing GMT therefore yields a negative ρ SIA.

Use an em dash for the sentence "Red boxes plot..." instead of a comma Negative sensitivity → positive sensitivity? Or is it largest magnitude sensitivity? When examining different fields - which fields?

Edits are added here (page 4, line 188):

Each of the ensembles reproduces the qualitative seasonal cycle of ρ SIA, with the largest magnitude sensitivity to warming from June to September.

and (page 2, line 121):

Because of the large difference between models in SIA and MIZF mean states, trends are expressed as a percentage change per year (this choice is examined in more detail in the Discussion below).

P3 MIZ sensitivity is rhoMIZF so shouldn't it be MIZF sensitivity

Yes, this is fixed here and throughout.

Start a new paragraph with "Despite producing..." because the topic has changed

Done!

How does inter-model spread explain MIZF sensitivity?

This sentence was removed as it was extraneous.

Section: Future projections of MIZ variability (MIZF variability?)

Fixed!

Section: Discussion: p2 This bias (a) is → this bias is (a).

I believe as the verb "is" mirrors the verb "can" after (b), it should stay.

P4 ...and those predict... → and those predicting

Fixed!

Data and Methods Which members are in the CMIP6-EME?

Please see Table 1 where all models and ensembles included in this manuscript are discussed.

Similarly there are only two. . . → This is obvious and doesn't need to be explicitly stated

Of course, but as this was an explicit question that came up when presenting, I prefer to leave it in.

Is the reason for a lack of CMIP6 future forcing scenarios of RCP8.5 because of the new projection pathways or because they don't provide enough future members under any scenario?

I now clarify in the text (page 7, line 471):

No CMIP6 models contributed 10 ensemble members under the same RCP8.5 forcing scenario as the CLIVAR-LE, and therefore are not included in future projections.

Sea ice observations of sea ice concentration - redundant

Fixed!

Switch between passive and active voice here

Edited as follows (page 7, line 484):

Pole holes in the NASAteam and Bootstrap products are filled by taking the average concentration around the perimeter of the pole hole as in *Olason and Notz* (2014). The OSISAF-provided pole hole filling is used for that product.

SIA is calculated as..→ multiplied the area of the cell, correct?

Indeed, this is added (page 8, line 486):

Sea ice area (SIA) is calculated as the sum of sea ice concentration multiplied by grid cell area in all grid cells above 0°N.

Notz (2014) → (Notz, 2014)

Fixed!

All supplemental figures are lacking captions

All have been edited in the revised manuscript.

Reviewer 3: Using model ensembles from CMIP6 and the CLIVAR-LE, this manuscript attempts to further understand the high sensitivity of Arctic sea ice area to warming in climate models. Findings show that for nearly all months, none of the models reasonably estimate the sea ice area sensitivity to warming, and that this is not a result of internal variability or inter-model differences. To combat this, a new metric, the MIZF (marginal ice zone fraction), is developed that is less sensitive to warming biases, and that can ultimately be used to compare sea ice model skill. The manuscript is sound, and the conclusions are appropriately supported by the figures/results. Overall this work represents a valuable contribution to the sea ice community (and the GCM community), and I recommend it for publication after the following minor comments/revisions are addressed.

Thank you for these valuable comments: all are addressed below in line and in the manuscript.

Minor comments: 1. In the introduction you say that the CMIP6-EME has 8 members but in the methods you say that it has 10 members. Later on in that part of the methods you explain that you at least left 1 model (MPI-HAM) out of the EME because it didn't have enough members. Was this the case for the second model that was left out? I think it would help if you add another sentence in the methods clarifying that the total number of members in the EME is 8.

Indeed, this was a typo - at the time of submission only 8 of the 20+ models with CMIP historical submissions had 10 ensemble members. A table now provides the list of models used, and 10 has been changed to 8 in the text.

2. Which 8 CMIP6 models are used in the EME? I am also curious about this because some CMIP6 models use the same sea ice and ocean models, and I wondered if you consider such models as independent or not when computing statistics for the EME.

Thanks, please see the list of included models in the Data and Method section. The issue of model interdependence is discussed on page 6, line 381:

Many of the sea-ice models used in modern GCMs share common parameterizations or development histories, and the interrelation of sea ice and ocean models could impact estimates of inter-model variability (*Knutti et al., 2017*). Yet developing a method to properly weight ensemble mean statistics based on climate model interdependence to form the CMIP-EME is not within the scope of this study, and I note that despite sharing the same sea ice and ocean model components, the two GFDL contributions to CLIVAR-LE diverge significantly in predictions of future MIZF.

3. There are a number of places in the text where parenthetical figure references could be used in lieu of lengthy figure explanations. This would also help shorten the text and/or free up space for further discussion. For example, in the section "Seasonal GCM estimates of SIA and MIZ sensitivities", the sentence that starts with "In Fig. 3. . ." could be removed and the next sentence describing the correlation results could parenthetically cite Fig. 3 instead. In some cases, the figure descriptions are also better suited to the captions. This is particularly true for parts of the description of Fig. 3 such as "Whiskers extend the boxes +/-3 standard deviations from the mean in each month, and are plotted against the observed annual cycle of SIA sensitivity (black), with observational uncertainty shown as +/-1 standard deviation of the observational mean."

This is a good point. These long figure descriptions of Fig 1 and 3 are now moved to captions.

4. Units of Kelvin do not have a degree symbol

Thanks for this, I did not know that! Fixed throughout and forevermore.

Minor figure suggestions: 1. Figure 1a: The black dashed line is somewhat difficult to distinguish in this panel. I wonder if a dotted line would be better here although it could very well make things worse. I hesitate to suggest adding another color to the plot but that might also be an option. If you already tried all of that feel free to ignore this.

I agree and in light of Reviewer 2's comments the CMIP-EME is plotted along with error bars as if it were a CLIVAR-LE model.

2. Figure 4 is missing a legend for the scatterplot color associated with each model

Yes - this has been added!

Line-by-line (but without line numbers so this is going to be a fun exercise in figuring out whether or not I give good directions. Good luck out there!):

Apologies! The submitted manuscript should have had line numbers and the revision does.

1.Beginning of paragraph 4 of the intro: “respnse” should be “response”

Fixed!

2.Final paragraph of intro beginning with “Across this set of 5+1 ensembles...”: You can probably just say 6 here even if there are two different types of ensembles.

While this comment is well taken, I hope this notation does serve a purpose in highlighting the that the CMIP-EME is a different sort of ensemble than the CLIVAR-LEs. I have retained the notation and hope the reviewer doesn't mind this deviant behavior.

Also please decapitalize the “s” in GCMS

Done!

3.Figure 1 caption: “observation” should be pluralized in “Black lines are observation.” Same with “Crosses are observation.”

Fixed!

4.Paragraph 2 of “GCM estimates of September Sea Ice and MIZ Change”: I suggest rewriting the 3rd sentence as “Box plots of Δ SIA for each model are given in Fig. 1(e). Red boxes indicate models that produce implausible (see methods) estimates of Δ SIA given the observations (black box labeled OBS, Fig. 1e), and blue those that produce plausible estimates.” Or even better, I suggest getting rid of most of this and making all of the references to Fig. 1 parenthetical as I mentioned in the minor comments section.

This has been amended to read (page 2, line 134):

While individual CLIVAR-LE models do not produce plausible estimates (see methods) of Δ SIA in September (red boxes, Fig. 1(e)), the CMIP-EME does (blue box, Fig. 1(e)), consistent with *Stroeve et al. (2012)*; *SIMIP Community (2020)*, as the observed September SIA trends are contained within the CMIP5/6 model ensemble spread.

5.And the next sentence “While individual CLIVAR-LE models do not individually produce...”: “individually” is redundant and can be omitted

Thanks! Fixed.

6.Paragraph 3 of “GCM estimates of September Sea Ice and MIZ Change”: I suggest rewriting the beginning of the first sentence as “the same analysis for marginal ice zone fraction (MIZF; Fig. 1b,d,f) reveals..”

Agreed, and done!

7.In the next sentence, “satellite” should be “satellites”

Fixed!

8. Figure 3 caption: 1. Second sentence: I suggest a minor rephrasing to “Boxes cover the interquartile range and horizontal whiskers the ensemble median.” 2. Later on there should be a “the” between “is” and “mean” in this sentence: “Black solid line is mean from observations.”

Thanks, both edits are included.

9. Discussion, 2nd paragraph, first sentence: you can get rid of “period from” in this

I removed this phrasing from all similar locations in the manuscript as it is indeed redundant.

10. Discussion, 3rd paragraph, sentence that starts with “Evaluating the trends of dimensional fields...” I have never heard the term “dimensional” although the internet tells me it’s a word haha. I still think dimensional is a better word here, but it’s your call

Dimensional does sound better, fixed!

11. Discussion, 3rd paragraph, last sentence: there is an extraneous “versus” in this sentence after “10 million sq km”

Fixed!

12. Discussion, 4th paragraph, 1st sentence, “predict” should be “predicting” to make it consistent with the earlier parts of the sentence

Fixed!

13. Last sentence of the 1st paragraph of the CMIP6 data section in the methods: “Internal variability expressed in the CMIP6-LE reflects model-to-model variability.” Should that be the CMIP6-EME?

Yes, and fixed, thank you!

14. Last sentence of the 2nd paragraph of the “uncertainty quantification” section of the methods: “extent” should be “extend”

Fixed, thank you!

REVIEWERS' COMMENTS

Reviewer #1 (Remarks to the Author):

I very much appreciate the detailed response to my comments from the first round of reviews and agree to publish the manuscript after considering a few minor comments.

(1) Abstract: "Global climate models (GCMs) consistently underestimate the loss of September Arctic sea-ice area, and exhibit a linear correlation between modeled sea-ice area losses and the magnitude of modeled global mean temperature increases."

Given Fig. 1 of the manuscript (and in addition e.g. fig 1(b) of SIMIP Community (2020)), I think the first part of this sentence is an overstatement and contradicts also with L335 " I find that trends in SIA alone are plausibly produced by the CMIP6-EME (Fig. 1e)". It is the sensitivity of SIA to warming which is not plausibly reproduced. Please, be more precise here.

(2) Fig 1 Caption: Shown in left part is SIA, not SIE, please correct.

(3) My former minor comment (8) was answered satisfactorily, with respective proposed changes in the text.

But unfortunately, that piece of new text was not included in the revised manuscript (right place should be L178). Please insert the revised text there.

Reviewer #2 (Remarks to the Author):

Summary:

I believe the significant revisions made by the author have made this manuscript much clearer. It is evident that extensive work has been done to clarify the methods, conclusions, and key points following the comments by the reviewers. I think the new title is great -- it much more clearly communicates the content of the paper and I find it very punchy! I am very satisfied with the many changes that have been made. I have just a couple very minor comments to add otherwise I believe this manuscript is fit for publication.

Page 2: line 55: a bit of a sentence fragment to finish off this sentence.

Page 2, line 88 and line 93: you use SIA sensitivity and MIZF sensitivity here but don't define what precisely is meant by that term in this paper until page 3, line 174.

Page 3, line 174-176: This sentence needs reworking. Either rhoSIA is meant to be in parentheses or maybe some commas are missing.

Page 6, line 361: a space between rho and MIZF has resulted in them being on two separate lines.

Reviewer #3 (Remarks to the Author):

All of my comments have been adequately addressed and I recommend this manuscript for publication.

A few minor things:

1. I like the new title, but maybe it could be "Marginal Ice Zone Fraction is a better benchmark than sea ice area for assessing sea ice and global climate model skill"? If this feels like too strong a claim then please feel free to keep the title as is. I understand that "better" could be pretty subjective here given that it depends on what particular aspect of GCM sea ice someone looks at, but at the same time the paper seems (to me at least) to argue that MIZF IS more useful because it isn't as strongly subject to model temperature biases.
2. Abstract: "I identify a new benchmark metric, marginal ice zone fraction (MIZF), which is less correlated..." Consider replacing 'less' with 'more weakly'
3. Line 85: I think there should be an 'after' between 'that' and 'accounting' here?
4. Line 110: "dashed" sold be "dotted"
5. Line 213: "predicting" should be "predict"
6. Line 271: there should be an "and" before "ensemble"
7. Throughout the discussion you use MIZF and sometimes MIZ fraction and sometimes you spell out the entire acronym again. It may be worthwhile to just stick with MIZF.
8. Lines 344-345: These references should be parenthetically cited
9. Line 380: "cleaner" may be a better word than "tighter" here
10. Line 434: remove "is ocean surface waves"

Thanks to the three reviewers again for their rapid review and comments, especially given how much stress everyone is under right now. All are now incorporated in the revised document. Reviewer comments are reproduced below in blue and as necessary, manuscript changes are provided in indented quotes.

Christopher Horvat

Reviewer 1:

very much appreciate the detailed response to my comments from the first round of reviews and agree to publish the manuscript after considering a few minor comments.

Thanks for your help in improving this work!

Abstract: “Global climate models (GCMs) consistently underestimate the loss of September Arctic sea-ice area, and exhibit a linear correlation between modeled sea-ice area losses and the magnitude of modeled global mean temperature increases.”

Given Fig. 1 of the manuscript (and in addition e.g. fig 1(b) of SIMIP Community (2020)), I think the first part of this sentence is an overstatement and contradicts also with L335 “I find that trends in SIA alone are plausibly produced by the CMIP6-EME (Fig. 1e)”. It is the sensitivity of SIA to warming which is not plausibly reproduced. Please, be more precise here.

I appreciate you pointing this out as it was an oversight. This now reads:

Global climate models (GCMs) consistently underestimate the response of September Arctic sea-ice area to warming,

Fig 1 Caption: Shown in left part is SIA, not SIE, please correct.

Fixed!

My former minor comment (8) was answered satisfactorily, with respective proposed changes in the text. But unfortunately, that piece of new text was not included in the revised manuscript (right place should be L178). Please insert the revised text there.

Indeed you are right! I re-added this into the text.

Reviewer 2:

I believe the significant revisions made by the author have made this manuscript much clearer. It is evident that extensive work has been done to clarify the methods, conclusions, and key points following the comments by the reviewers. I think the new title is great – it much more clearly communicates the content of the paper and I find it very punchy! I am very satisfied with the many changes that have been made. I have just a couple very minor comments to add otherwise I believe this manuscript is fit for publication.

Thanks very much for helping improve this work.

Page 2: line 55: a bit of a sentence fragment to finish off this sentence.

These two sentences were slightly altered for better tense and grammar.

The marginal ice zone (MIZ) has been a point of focus as a critical region for polar climate and ecology (*Wadhams et al.*, 1981; *Lowry et al.*, 2014; *Zippel and Thomson*, 2016), and can be observed using the same passive microwave observations used to estimate SIA/SIE (*Stroeve et al.*, 2016). MIZ location and variability is set through the coupled interaction of oceanic, atmospheric, thermodynamic and sea ice dynamical processes that are focus of many recent modeling efforts (*Williams et al.*, 2013; *Horvat and Tziperman*, 2015; *Manucharyan and Thompson*, 2017; *Wenta and Herman*, 2018; *Roach et al.*, 2018).

Page 2, line 88 and line 93: you use SIA sensitivity and MIZF sensitivity here but don't define what precisely is meant by that term in this paper until page 3, line 174.

Good point. I amended this sentence to read:

Across this set of 5+1 ensembles, I consider two main quantities: SIA and MIZF, and their sensitivities to warming - the relationship between a unit change in SIA/MIZF and a unit change in GMT.

Page 3, line 174-176: This sentence needs reworking. Either rhoSIA is meant to be in parentheses or maybe some commas are missing.

This sentence was re-written:

Here I investigate this correlation, examining SIA and MIZF sensitivity to warming across the seasonal cycle, where the sea ice sensitivity (or MIZF sensitivity), ρ SIA, is defined as the percent change in SIA per degree change in GMT (units %/K).

Page 6, line 361: a space between rho and MIZF has resulted in them being on two separate lines.

Fixed!

Reviewer 3:

All of my comments have been adequately addressed and I recommend this manuscript for publication.

Thanks very much for your help revising the manuscript!

A few minor things:

1. I like the new title, but maybe it could be "Marginal Ice Zone Fraction is a better benchmark than sea ice area for assessing sea ice and global climate model skill"? If this feels like too strong a claim then please feel free to keep the title as is. I understand that "better" could be pretty subjective here given that it depends on what particular aspect of GCM sea ice someone looks at, but at the same time the paper seems (to me at least) to argue that MIZF IS more useful because it isn't as strongly subject to model temperature biases.

I struggled with this as you could tell, though there is a 15 word limit on titles. Thus I went to:

Marginal Ice Zone Fraction: a Better Climate Model Benchmark than Sea Ice Area.

Abstract: “I identify a new benchmark metric, marginal ice zone fraction (MIZF), which is less correlated. . .” Consider replacing ‘less’ with ‘more weakly’

I think this is an unfortunate case of English not giving us a good word. I kept “less” to avoid the double comparative.

Line 85: I think there should be an ‘after’ between ‘that’ and ‘accounting’ here?

Added!

Line 110: “dashed” sold be “dotted”

Fixed!

Line 213: “predicting” should be “predict”

Fixed!

Line 271: there should be an “and” before “ensemble”

Good catch! Fixed.

Throughout the discussion you use MIZF and sometimes MIZ fraction and sometimes you spell out the entire acronym again. It may be worthwhile to just stick with MIZF.

I have now removed all MIZ fractions after MIZF is defined.

Lines 344-345: These references should be parenthetically cited

Fixed!

Line 380: “cleaner” may be a better word than “tighter” here

Fixed!

Line 434: remove “is ocean surface waves”

Fixed!